METHODS

# IonBench: A benchmark of optimisation strategies for mathematical models of ion channel currents

**Matt J. Owen** ID ¤, **Gary R. Mirams** ID *

Centre for Mathematical Medicine and Biology, School of Mathematical Sciences, University of Nottingham, Nottingham, United Kingdom

¤ Current address: School of Biological Sciences, University of Bristol, Bristol, United Kingdom
* gary.mirams@nottingham.ac.uk

**Data availability statement:** The most recent version of ionBench can be found on GitHub

## Abstract

Ion channel models present many challenging optimisation problems. These include unidentifiable parameters, noisy data, unobserved states, and a combination of both fast and slow timescales. This can make it difficult to choose a suitable optimisation routine *a priori*. Nevertheless, many attempts have been made to design optimisation routines specifically for ion channel models, however, little work has been done to compare these optimisation approaches. We have developed ionBench, an open-source optimisation benchmarking framework, to evaluate and compare these approaches against a standard set of ion channel optimisation problems. We included implementations of thirty-four unique optimisation approaches that have been previously applied to ion channel models and evaluated them against the ionBench test suite, consisting of five parameter optimisation problems derived from the cardiac ion channel literature. Each optimisation approach was initiated from multiple starting parameters and tasked with reproducing a problem-specific simulated dataset. Through ionBench, we tracked and evaluated the performance of these optimisations, identifying the expected run time until a successful optimisation for each approach, which was used for comparisons. Finally, we used these results, in addition to other literature results, to identify a new efficient approach. Its use could reduce computation time by multiple orders of magnitude, while also improving the reliability of ion channel parameter optimisation.

## Author summary

To work with mathematical models of ion channels, model parameters must be configured to ensure the model can reproduce experimental data. This is achieved through a process called parameter optimisation. Many different methods and approaches have been developed to optimise parameters in ion channel models and they can vary in both speed (how fast they can optimise) and efficacy (how likely they are to produce

https://github.com/CardiacModelling/ionBench. An archived version of the code at time of publication and all generated optimisation data are available on Zenodo (https://doi.org/10.5281/zenodo.15754015 and https://doi.org/10.5281/zenodo.14072634, respectively).

**Funding:** This work was supported by the Wellcome Trust (https://wellcome.org, grant no. 212203/Z/18/Z) via a Senior Research Fellowship to GRM. This research was funded in whole, or in part, by the Wellcome Trust [212203/Z/18/Z]. For the purpose of open access, the author has applied a CC-BY public copyright licence to any Author Accepted Manuscript version arising from this submission. The funders had no role in study design, data collection and analysis, decision to publish, or preparation of the manuscript.

**Competing interests:** The authors have declared that no competing interests exist.

parameters that can reproduce the data). However, even though this variability is known, a lack of standard problems has meant we still do not know the best way to optimise these models. We have compared thirty-four unique optimisation approaches that have been previously applied to ion channel models against a set of five optimisation problems derived from the literature. Finally, we used these results, in addition to other literature results, to identify a new efficient approach. Using this approach can lead to significant savings in computational resources and improvements in the reliability of parameter optimisation for ion channel models.

## Introduction

Many mathematical models in systems biology require calibrating to experimental data via parameter optimisation. Mathematical models of ion channels are no exception to this. In particular, we generally only observe a current, which is an indirect measurement of (typically) a single conformational state of the channel, the open state; while the remaining inactive or closed states are unobservable. In the most common Markov model ordinary differential equation (ODE) framework for channel modelling, these conformational states are also state variables in the ODE system. Since many parameters cover transitions from these unobserved states, inferring their values using a mathematical model has proven to be an effective technique for learning these rates.

Calibrating a model to data via parameter optimisation is an important part of constructing an ion channel model, and many optimisation approaches have been applied to ion channel models. However, these methods vary in both effectiveness [1,2] and speed [3] and little work has been done to explore these different choices and give recommendations on their usage. Vanier & Bower [2] compared four parameter optimisation methods (and one simple Monte Carlo method as a reference) against one another for five neuron action potential model optimisation problems for conductance and channel time constant parameters and found a simulated annealing method to perform best. However, the large number of different methods that have been used for ion channel models (Fig 1) make results from this relatively small collection of methods difficult to generalise.

The standard tool for handling this kind of problem is benchmarking: developing a set of standard problems against which you can test a wide variety of optimisation approaches. From these results, more general recommendations and guidance can be derived to assist in the development and parameterisation of future models [4].

Many of the optimizers currently used are gradient-free optimisers, which may be necessary in highly multi-modal objective functions [5]. However, a previous systems biology optimisation benchmark suggested favouring the use of gradient information when optimising ODE models due to the availability of sensitivity equations to calculate the gradient [6]. They also found local optimisers, utilising a multi-start approach (or other global meta-heuristic) to outperform global optimisers, although this conclusion is not universal across other studies [7,8].

While no large-scale benchmarks have been developed for ion channel model optimisation, more general optimisation benchmarks do exist. For example, the optimisation benchmarking framework COCO [9] contains a wide variety of other types of benchmark problems. A large scale benchmark comparing 31 optimisers suggests that the optimal choice of optimiser depends highly on the computational budget and number of parameter dimensions [8]. Such works have developed a range of criteria and performance metrics for evaluating

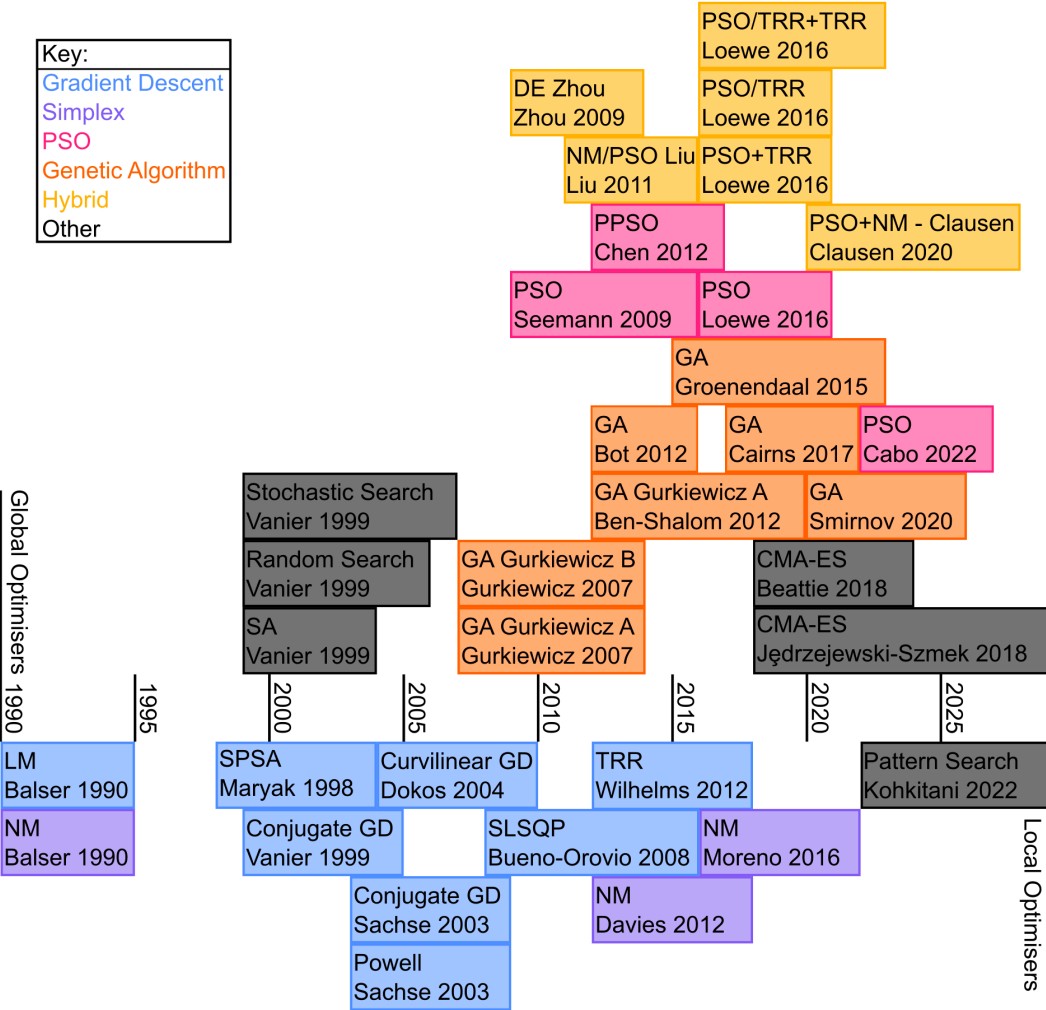

**Fig 1. Approach timeline.** Different optimisation approaches applied to ion channel models over time. Each cell gives the name of the optimisation algorithm, followed by the surname of the first author and year of publication.

optimisation. However, it remains unclear if the results from any of these benchmarks generalise to ion channel model optimisation.

In contrast to the broad range of models in systems biology, ion channel optimisation is very consistent in problem definition. For example, it is rare to see ion channel optimisation with only 1 or 2 parameters and equally rare to see an optimisation of more than 100 parameters. It is also common to see problems occurring in terms of parameter identifiability and information content [10,11] or ODE solver tolerances and noise [12,13]. We therefore concluded that a set of ion channel specific benchmark problems would be useful to compare these optimisation approaches.

Here, we compare a wide range of optimisation methods that have been previously applied to ion channels. We investigate their performance against a set of standardised problems from the literature and derive recommendations on their usage. Finally, we utilise these comparisons, alongside other recommendations from the literature, to derive a new efficient optimisation approach.

Additionally, we made this code, named `ionBench`, open source to allow others to evaluate performance of future optimisation approaches, and to add new benchmark problems. Hopefully, this enables more efficient optimisation methods to be developed and utilised in future.

## Materials and methods

### Terminology

We begin with a description of some key terms used in `ionBench`. These distinguish between aspects and components of the *optimisation problem* compared with the *optimisation approach*. These key terms and their relationships are demonstrated diagrammatically in Fig 2 and also outlined below.

**Problems:** *Problems* define the test suite in `ionBench`. From the point of view of the optimisation, these are predominantly black boxes describing a *cost function* which evaluates the quality of a parameter vector by simulating model output with this parameter vector and returning some measure of distance between the output and data. A problem is predominantly defined by a model, protocol(s)/experiment(s) to perform, a method to randomly sample parameters for initial guesses, output data and a cost function, e.g. sum of squared errors. Most importantly, none of these aspects of the problem should vary between different approaches. For example, the sampling function is not dependent on parameter transforms defined in the modification/approach.

**Optimiser:** An *optimiser* is the general algorithm which searches parameter space to minimise a cost function. It should only need access to problem-specific functions to sample parameters and to evaluate them with the problem's cost/objective function, either solved for the cost or the gradient (chosen by the optimiser). Again, no details of the optimiser, other than which of these functions it uses, should change for the different problems. Examples include Nelder-Mead [14], Trust Region Reflective (TRR) [15], Covariance Matrix Adaptation Evolution Strategy (CMA-ES) [16], or Particle Swarm Optimisation (PSO) [17].

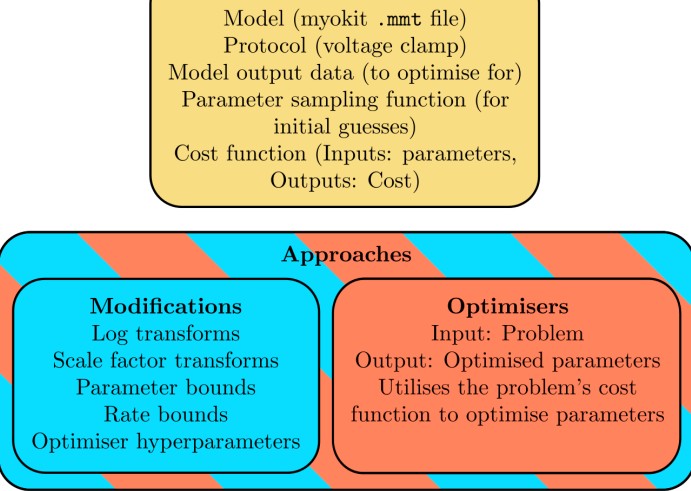

**Fig 2. Keywords.** A visual representation of the four key terms used in `ionBench`: problem, approach, modification, and optimiser.

**Modification:** The *modification* acts as an interface between the optimiser and the problem as a way to allow more flexibility and control over the optimisation. For example, two papers may both use CMA-ES, but one may enforce positivity of a parameter by searching in a log-transformed space while the other utilises bounds. These kind of differences are handled by the modification. For the sake of standardisation, they are limited to log-transforms, scale factor transformations, applying bounds on each of the parameters, and applying additional bounds on model transition rates (functions of multiple parameters). These modifications are implemented as binary on/off settings applied in a standard problem-specific form. For example, applying bounds will use the problem-specific parameter sampling bounds, or applying log-transforms will apply log-transforms to a predefined problem-specific set of parameters. Additionally, we include optimiser hyperparameters (such as the number of particles in an optimiser's swarm) as part of the modification.

**Approach:** An *approach* is a pairing between an optimiser and a modification. From a description of a specific ion channel optimisation in the literature, we extract the optimiser and modifications to label as the approach used. This means some information in the original descriptions of these approaches is not carried over to `ionBench`; for example, if a paper suggests a different cost function, this is not included since it is not an optimiser or a modification setting. However, this choice ensures results from different approaches applied to different problems are easily comparable.

Optimisers are organised into six categories: Gradient Descent, Simplex, Genetic Algorithms, PSO, Hybrid, and Other. Hybrid methods utilise the following notation:

- Opt1+Opt2 denotes running the optimiser Opt1 and then, once it has terminated, running optimiser Opt2;
- Opt1/Opt2 denotes running optimiser 1 and optimiser 2 concurrently in some way, typically running optimiser Opt2 every *n* iterations of optimiser Opt1.

While the separation between 'problem' and 'approach' is somewhat arbitrary, the above definitions should help to eliminate ambiguity is this separation. For example, one may ask whether parameter transforms influence the parameter sampling functions (does applying log-transforms mean that optimizers internally sample parameters log-uniformly rather than uniformly). Since the parameter sampling function is part of the problem and the transforms are part of the modification (which itself is part of the approach), and the definition of all aspects of the problem cannot be influenced by the approach, then it is clear that the sampling function distribution does not change depending on parameter transforms. We will see later why we do not want the approach to influence the parameter sampling function.

## Requirements for approaches

To ensure comparability between different published approaches, it is necessary to place some restrictions on which are included into `ionBench`. We have aimed to include the majority of ion channel optimisation approaches, while ensuring comparisons between the approaches remains 'fair'. This means approaches should require similar amounts of information as inputs (an approach that requires knowing the model structure *a priori* should not be compared with one that works to identify it) and similar amounts of information as outputs (Bayesian methods that approximate samples of a full posterior distribution should not be compared with methods deriving a single point estimate). These restrictions are given below.

- Only parameter optimisations are included, not approaches that are designed to optimise the model structure.

- We do not include Bayesian methods (e.g. approximate Bayesian computation, Markov chain Monte Carlo, or variational inference).
- We do not include methods which require pre-trained algorithms [18].
- We do not include multi-objective optimisations (such as identifying Pareto fronts).
- While we focus on approaches applied to cardiac ion channel models, we do consider those applied to other ion channel models where possible.
- We do not include optimisation approaches for stochastic models, such as single ion channel models.

We searched the literature for work which included optimisation of ion channel models, identifying 30 papers that met these requirements, defining a total of 42 approaches, 34 of which were unique. Table 1 presents a summary of these 42 optimisation approaches, including naming each approach, the optimisers used, and the approach-specific modifications.

## Structure of ionBench

Our new Python package `ionBench` performs parameter optimisation on a standard set of cardiac ion channel optimisation problems derived from the literature. Each of the implemented optimisation problems can be treated as a black box, to which parameters can be proposed and a cost (or alternatively, the gradient of the cost function) will be returned to allow optimisation. Each time the optimiser proposes new parameters, `ionBench` tracks and stores information on the current state of the optimisation to evaluate its performance. A description of how the models' equations are solved is given in the section Forward solvers.

In addition to tracking externally-implemented optimisation approaches, `ionBench` also contains implementations of the 42 optimisation approaches given in Table 1 (a mix of wrappers for common standard optimisers, such as Nelder-Mead or CMA-ES, and new implementations of other optimisers where common standard implementations are unavailable). A detailed description of the implementation of `ionBench` is available in the form of a series of `README.md` files with the code (see Data Availability).

## Problems

A good test suite of problems is one that efficiently covers the range of challenges faced. This way, if an approach performs well on a subset of the problems, we can infer what properties contribute to its performance, and this information can guide the choice of optimisation approach in future work. In the case of cardiac ion channel modelling, this means:

- capturing ion channels of different speeds ($I_{Kr}$, $I_{Kur}$, $I_{Na}$),
- including both Hodgkin-Huxley and Markov model formulations,
- utlising cost functions defined on both the current trace and on derived summary statistics,
- using data both with and without noise (while real data always contains noise, optimisation approaches are commonly evaluated without noise),
- a range of different numbers of parameters for optimisation (around 5 to 25),
- different sizes of parameter search spaces, and
- both high information and low information protocols.

**Table 1. Optimisation approaches.**

| Name [Citation] | Optimiser | Gradients | Transforms | | Bounds | |
|---|---|---|---|---|---|---|
| | | | Log | Scaling | Parameter | Rate |
| Balser1990a [19] | GD (LM) | ✓ | | | | |
| Balser1990b [19] | Simplex (NM) | | | | | |
| Maryak1998 [20] | GD (SPSA) | ✓ | | | | |
| Clancy1999 [21] | GD (LM) | ✓ | | | | |
| Vanier1999a [2] | GD (Conjugate) | ✓ | | ✓ | ✓ | |
| Vanier1999b [2] | Other (Simulated Annealing) | | | ✓ | ✓ | |
| Vanier1999c [2] | Other (Stochastic Search) | | | ✓ | ✓ | |
| Vanier1999d [2] | Other (Random Search) | | | ✓ | ✓ | |
| Sachse2003a [22] | GD (Conjugate) | ✓ | | | | |
| Sachse2003b [22] | GD (Powell's) | ✓ | | | | |
| Dokos2004 [23] | GD (Curvilinear) | ✓ | | | ✓ | |
| Gurkiewicz2007a [24] | GA | | | | ✓ | |
| Gurkiewicz2007b [24] | GA | | | | ✓ | |
| Bueno-Orovio2008 [25] | GD (SQP) | ✓ | | | ✓ | |
| Seemann2009a [26] | PSO | | | | | |
| Seemann2009b [26] | GD (Powell's) | ✓ | | | | |
| Zhou2009 [27] | Hybrid (DE/LM) | ✓ | | | | |
| Guo2010 [28] | GD (Curvilinear) | ✓ | | | ✓ | |
| Liu2011 [29] | Hybrid (PSO/NM) | | | | ✓ | |
| Ben-Shalom2012 [30] | GA | | | | ✓ | |
| Bot2012 [31] | GA | | | ✓ | ✓ | |
| Chen2012 [32] | PSO | | | | ✓ | |
| Davies2012 [33] | Simplex (NM) | | | ✓ | ✓ | |
| Wilhelms2012a [34] | GD (Powell's) | ✓ | | | | |
| Wilhelms2012b [34] | GD (TRR) | ✓ | | | ✓ | |
| Abed2013 [35] | GD (Curvilinear) | ✓ | | | ✓ | |
| Du2014 [36] | GD (TRR) | ✓ | | | ✓ | |
| Groenendaal2015 [37] | GA | | | ✓ | ✓ | |
| Loewe2016a [3] | GD (TRR) | ✓ | | | ✓ | |
| Loewe2016b [3] | PSO | | | | ✓ | |
| Loewe2016c [3] | Hybrid (PSO+TRR) | ✓ | | | ✓ | |
| Loewe2016d [3] | Hybrid (PSO/TRR) | ✓ | | | ✓ | |
| Loewe2016e [3] | Hybrid (PSO/TRR+TRR) | ✓ | | | ✓ | |
| Moreno2016 [38] | Simplex (NM) | | | | ✓ | |
| Cairns2017 [39] | GA | | | | ✓ | |
| Jędrzejewski-Szmek2018 [40] | Other (CMA-ES) | | | | ✓ | |
| Beattie2018 [41] | Other (CMA-ES) | | ✓ | | ✓ | ✓ |
| Clerx2019 [12] | Other (CMA-ES) | | ✓ | | ✓ | ✓ |
| Clausen2020 [42] | Hybrid (PSO+NM) | | | | ✓ | |
| Smirnov2020 [1] | GA | | | ✓ | ✓ | |
| Cabo2022 [43] | PSO | | | | ✓ | |
| Kohjitani2022 [44] | Other (Pattern Search) | | | ✓ | | |

Each approach is named by first authors surname and year of publication, followed by an a-z identifier if multiple approaches were published. Each approach lists the optimiser used, whether the optimiser makes use of gradient information, whether log or scaling factor transforms were used on parameters, and whether bounds on parameters or rates were used. Approaches that are duplicates of a previously published approach (either directly citing a previous approach or describing an identical approach independently) are given a shaded background. LM: Levenberg Marquardt, NM: Nelder Mead, SPSA: Simultaneous Perturbation Stochastic Approximation, SQP: Sequential Quadratic Programming, DE: Differential Evolution.

We identified and derived a test suite of five problems which were then implemented in `ionBench`. Of these five problems, three were used exactly as given in the literature, and two were created based on a handful of works.

The first two problems (those not taken directly from the literature) are named 'Staircase HH' (Hodgkin-Huxley) and 'Staircase MM' (Markov Model). Both of these problems use the staircase voltage protocol [45] and they simulate $I_{Kr}$ with a Hodgkin-Huxley model [41] and a Markov model [46], respectively. These problems then have additional noise added to the current trace, forming the data for the optimisation, as described in Sect A.1 in S1 Supporting Information. The parameter sampling region utilises rate bounds and log transforms, the most efficient problem definition in Clerx et al. [12]. The current trace and voltage protocols are shown in Fig 3, the model structures are given in Fig 4A–4B, and the parameter sampling regions and rate bounds are demonstrated in Fig 5A.

The next two problems are named 'Loewe $I_{Kr}$' and 'Loewe $I_{Kur}$' [3]. These include large $I_{Kr}$ and $I_{Kur}$ Hodgkin-Huxley models [47] (Fig 4C–4D) and a simple step protocol on which the current trace is recorded to form the problem data (Fig 3). The parameters are separated into two groups, multiplicative (those that appear as a scaling factor in the rates) or additive (those that appear as $\pm$ offsets in the rates). The parameter sampling region samples multiplicative parameters log-uniformly between $\times 0.1$ and $\times 10$ the true parameters and additive parameters uniformly $\pm 60$ around the true parameters.

The final problem is named 'Moreno $I_{Na}$' and uses an $I_{Na}$ Markov model (Fig 4E) [38]. The data comprise summary statistics (a subset of the full summary statistics originally used [38]) measuring steady state inactivation, steady state activation, recovery from inhibition, and time to 50% decay of current, demonstrated in Fig 3. The parameter sampling region is defined as $\pm 25\%$ around the true parameters.

Table 2 summarises these models in the context of the challenges outlined above. This set of 5 problems achieves good coverage over each of these challenges, ensuring that if an approach only performs well for some challenge properties and not others, we should be able to identify which challenges affect performance. A detailed description of each of these problems and their implementation is given in S1 Supporting Information, including solver tolerances (Sect A.5 in S1 Supporting Information), parameter values (Tables A–E in S1 Supporting Information), initial conditions (Sect A.4 in S1 Supporting Information), and cost functions (Sect A.2 in S1 Supporting Information).

## Forward solvers

Each of the models' equations (Fig 4) are required to be solved whenever the cost at a new proposed parameter set is evaluated. All solvers are implemented in `myokit` [48], with the Staircase problems utilising CVODE [49,50], the Loewe problems utilising an analytic solution (as used for HH gating variables in the Rush-Larsen scheme [51]), and the Moreno $I_{Na}$ problem using an analytic linear model solver.

To calculate the gradient of the cost function, we require the model to be solved with sensitivities (this increases computation time but provides more information about the cost function). While it is possible to differentiate the analytical solutions with respect to model parameters, it is complex to do so across voltage steps. As such, sensitivities are not available in `myokit` for analytically solved models. We use the CVODES solver [52] in `myokit` for this, across all problems.

For CVODE and CVODES, solver tolerances are required. We choose solver tolerances to ensure the magnitude of the ODE solver noise is below a series of thresholds [13]. The specific

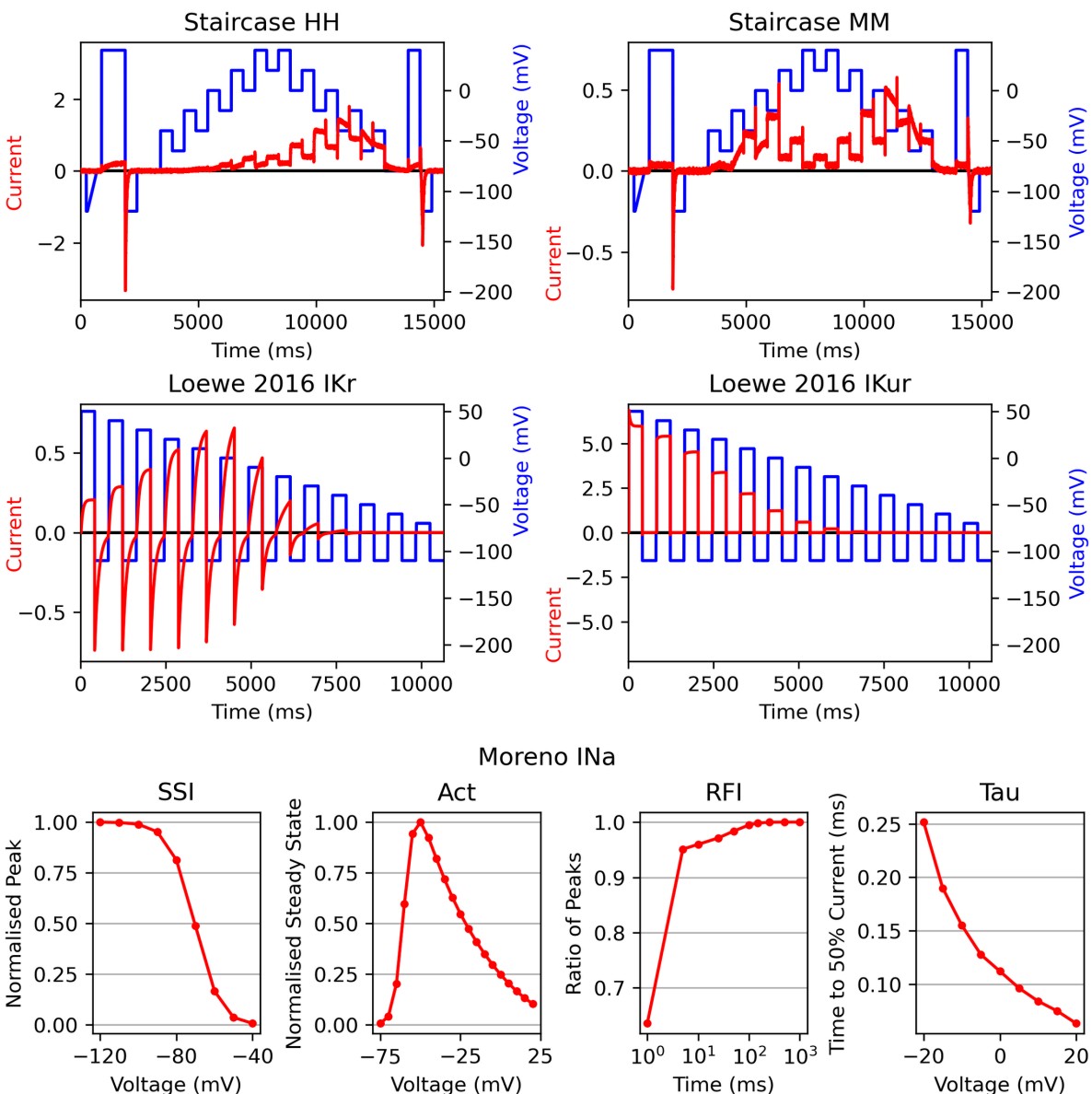

**Fig 3. Optimisation problems: Data.** The simulated data for each of the problems. For the Staircase and Loewe problems, the voltage protocol (blue) and the current data (red) are given. For the Moreno problem, the four summary statistic curves that are used in `ionBench` are reported.

requirements for the solver tolerances, as well as the tolerances that are used in this work, are provided in Sect A.5 in S1 Supporting Information.

## Approaches and optimisers

The 34 unique approaches shown in Table 1 require an implementation in `ionBench` before their performance can be evaluated. These optimisers are implemented using `scipy` [53], `pints` [54] or `pymoo` [55] where possible. If no implementation of the optimiser could be

**Fig 4. Optimisation problems: Models.** The models underlying each of the problems in `ionBench`. Both the model structure and model equations and rates are reported for the: (A) Staircase HH; (B) Staircase MM; (C) Loewe $I_{Kr}$; (D) Loewe $I_{Kur}$; and (E) Moreno $I_{Na}$ problems.

found in the original paper or a standard Python package, we have developed a Python implementation which is included in `ionBench`. If the optimiser is insufficiently described to produce an implementation, such as missing information on initialisation routines or hyperparameter values, we have included an optimiser which is consistent with all reported information. Further details on each of the approaches are given in the `README.md` files supplied with the code (see Data Availability).

## Modifications

As summarised previously, modifications are a list of settings that can be enabled for an approach. These settings are outlined in detail below.

- Log transforms: Each problem has a set of parameters that will be log transformed if this setting is enabled. Which specific parameters are log transformed when this setting is enabled are given for each problem in Tables A–E in S1 Supporting Information. An approach is considered to use log transforms if any parameters were log transformed when the approach was applied.

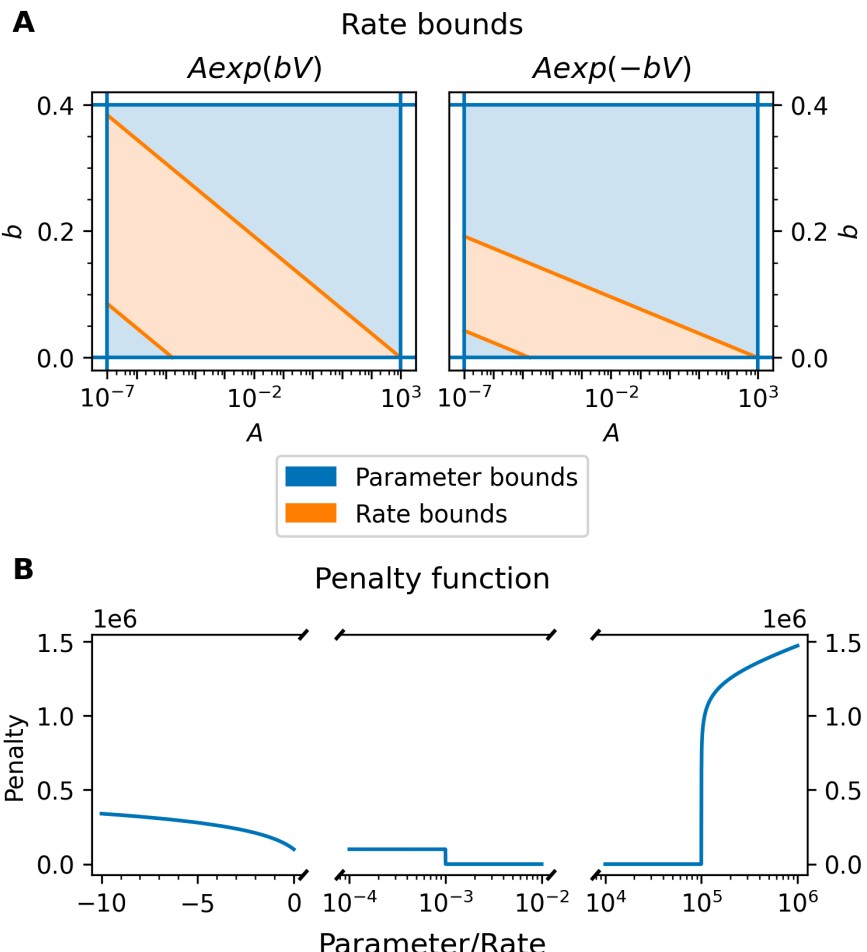

**Fig 5. Rate bounds and penalty function.** (A) The rate bounds for the two most common types of rates, $A \exp(bV)$ and $A \exp(-bV)$ [12,41]. The rate bounds specify upper and lower bounds that ensure the timescale for the maximum transition rate ($V \in [-120\,\text{mV}, 60\,\text{mV}]$) is between 1 µs and 1 min. For the Staircase problems, sampled parameters satisfy both the parameter bounds and the rate bounds (orange region). (B) An example plot of the penalty function. The lower bound is at $10^{-3}$ while the upper bound is at $10^5$.

- Scale factor transforms: If scale factor transforms are enabled, all parameters are rescaled around the true values (such that one is the true value for all parameters). An approach is considered to use scale factor transforms if any parameters were linearly rescaled when the approach was applied.
- Parameter bounds: If parameter bounds are enabled, upper and lower bounds are enforced on all parameters. These bounds are the same as those that define the parameter sampling region for each problem. If parameters outside of these bounds are attempted to be solved by the optimiser, instead of solving the model, a penalty function is applied. This setting is enabled for an approach/modification if any parameters were bounded in the original description of the approach.
- Rate bounds: These define upper and lower bounds on the rates (functions of parameters) to ensure model transitions happen on a reasonable timescale (bounded between 1 min and 1 µs, giving rates between $1.67 \times 10^{-5}\,\text{ms}^{-1}$ and $1000\,\text{ms}^{-1}$). If parameters that violate these bounds are attempted, they again use the penalty function instead of

**Table 2. Optimisation challenges.**

| Challenge | Staircase HH | Staircase MM | Loewe $I_{Kr}$ | Loewe $I_{Kur}$ | Moreno $I_{Na}$ |
|---|---|---|---|---|---|
| Ion channel | $I_{Kr}$ | $I_{Kr}$ | $I_{Kr}$ | $I_{Kur}$ | $I_{Na}$ |
| Model formulation | Hodgkin-Huxley | Markov | Hodgkin-Huxley | Hodgkin-Huxley | Markov |
| Cost function definition | Current trace | Current trace | Current trace | Current trace | Summary statistics |
| Noise | Yes | Yes | No | No | No |
| Number of parameters | 9 | 15 | 12 | 25 | 16 |
| Parameter search space | Large | Large | Medium | Medium | Small |
| Protocol information | High | High | Low | Low | Low |

The properties of each model for each of the challenges in cardiac model optimisation described previously.

attempting to solve the model. This setting is enabled for any approach where parameter combinations were bounded (in a way different to rectangular $(p_1^{lb}, p_1^{ub}) \times (p_2^{lb}, p_2^{ub})$ bounds).

The parameter sampling region given by the Staircase problems is very broad and points outside of it are likely to result in a failed model simulation. To avoid this problem, the Staircase problems always have both parameter bounds and rate bounds enabled, regardless of the modification and approach.

The modification settings for each approach are given in Table 1. Optimiser hyperparameters (defaults and approach-specific changes) are available in the code.

## Parameter penalty function

The penalty function that is applied during bound violations is given by Eq (1), where:

- $H(x)$ is the Heaviside function ($H(x) = 1$ if $x \geqslant 0$, else $H(x) = 0$);
- $p_i$ is the $i$th parameter value;
- $p_i^{lb}$ and $p_i^{ub}$ are the lower and upper bounds of the $i$th parameter, respectively;
- $r_j(p)$ is the $j$th maximum transition rate, evaluated over $V \in [-120\,\text{mV}, 60\,\text{mV}]$, for the parameters $p$;
- $r_j^{lb}$ and $r_j^{ub}$ are the lower and upper bounds on the $j$th model transition rate.

$$\begin{aligned}
\text{Pen}(p) = 10^5 \sum_i \Big[ & H(p_i^{lb} - p_i)(1 + \ln(1 + p_i^{lb} - p_i)) \\
& + H(p_i - p_i^{ub})(1 + \ln(1 + p_i - p_i^{ub})) \Big] \\
+ 10^5 \sum_j \Big[ & H(r_j^{lb} - r_j(p))(1 + \ln(1 + r_j^{lb} - r_j(p))) \\
& + H(r_j(p) - r_j^{ub})(1 + \ln(1 + r_j(p) - r_j^{ub})) \Big].
\end{aligned} \tag{1}$$

This penalty function ensures no penalty is applied for any parameters that lie inside the bounds, then jumps discontinuously at the bounds. The penalty then increases approximately logarithmically for violations in large bounds (like the upper bounds on the rates), while also

increasing approximately linearly for violations in small bounds (like the lower bounds on rates) to allow for possibly negative rates to be assigned a finite penalty. A plot of a slice of the penalty function is shown in Fig 5B, where a violation in the upper bound from $10^5$ to $10^6$ produces a penalty on a similar scale to a violation in the lower bound from $10^{-3}$ to $-10$.

The values for $r_j^{lb}$ and $r_j^{ub}$ are fixed at $1.67 \times 10^{-5}$ ms$^{-1}$ and $1 \times 10^3$ ms$^{-1}$ in most cases. The only exception to this is in the Moreno $I_{Na}$ problem, where two of the rates violate these bounds significantly at the true parameters. For these two rates only, we increase $r_j^{ub}$ to $1 \times 10^7$ ms$^{-1}$. Transition rates on this sub-ns timescale are unlikely to be meaningfully identified on data recorded at ms or μs frequencies. However, for this benchmark, we need to ensure that the true parameters (defined in [38]) are accessible for all approaches.

## Optimisation

For each problem, we sample $n_{Run}$ parameter vectors ($n_{Run}$ = 10 for Staircase HH, Loewe $I_{Kr}$, and Moreno $I_{Na}$; $n_{Run}$ = 50 for Staircase MM; and $n_{Run}$ = 100 for Loewe $I_{Kur}$). We then begin $n_{Run}$ optimisations with each approach starting at each of these parameter vectors. Since the starting parameter vectors do not vary between the different approaches, this is the reason it is not possible to have parameter transforms influence the sampling functions of the problems.

While each optimisation is running, we use `ionBench` to track the following information:

- Each parameter vector the optimiser attempts;
- Whether that attempt was inside or outside any parameter or rate bounds;
- If the model was solved (i.e. if the parameters were inside the bounds), then we record whether the model was solved with or without sensitivities (solving the models with sensitivities is more expensive, so tracked separately);
- The time it took to solve the model;
- The cost at the attempted parameters, including the penalty, if applicable;
- The best cost seen so far, and the corresponding parameters.

An optimisation is considered successful if it identifies a parameter vector whose cost is below a problem-specific cost threshold. Otherwise, that optimisation is considered unsuccessful. Once an optimiser reaches the cost threshold, it terminates the optimisation.

We chose to use a cost threshold as the success criterion rather than parameter distance due to the possibility of having unidentifiable parameters in the models, which makes defining a suitable parameter distance challenging. We could have chosen to exclude any models that are unidentifiable. However, unidentifiability is a common challenging in cardiac ion channel optimisation [10], so excluding such models would limit the applicability of the benchmark results.

This cost threshold is calculated using profile likelihood plots generated for each problem. The cost threshold is given as the minimum cost displayed on the profile likelihood plots at the ±5% perturbation, one-at-a-time, around the true parameters, ignoring any parameters that showed unidentifiability or a strong noise bias in the global minimum. Further details, including which parameters were considered unidentifiable are reported in Sect D in S1 Supporting Information.

## Termination criteria

While some of the optimisers provide a range of termination criteria, others may only include a maximum number of iterations. While this is a useful termination criteria, setting the

maximum number of iterations is highly problem specific and can rarely be simply set *a priori*. This makes it challenging to include into a benchmark. Using a value that is too large will result in excess computation, penalising an approach which may have already converged, and too small a value could limit an approach from converging at all. For this reason, we set the maximum number of iterations very large ($2.5 \times 10^4$, or $10^4$ for genetic algorithms) across all approaches to discourage this type of termination. To ensure optimisers which rely on this are still able to terminate, we then apply two additional termination criteria to all optimiser runs.

The first termination criteria is a cost threshold. Once an optimisation reaches the desired cost threshold, it is considered to have succeeded and optimisation is terminated. The second termination criteria is used to detect when an optimisation has converged, but is unsuccessful. If the best cost found so far has not improved by $10^{-7}$ over the past 2500 model solves (not including out of bounds parameters), then the optimisation is considered to have converged unsuccessfully and so is terminated. An optimisation is considered finished once either of these termination criteria are triggered (which are checked every function evaluation), or when an optimiser-specific termination criteria is triggered (which may or may not be checked every function evaluation).

Additionally, we impose a hard limit on computation time for the optimisations. For each approach-problem pair, it is required to complete its $n_{\text{Run}}$ optimisations in 7 days (on one CPU core). If any pair exceeds these limits, it is deemed too expensive and excluded from the analysis.

## Performance metrics

To compare the optimisation results for each approach, we calculate the Expected Run Time (ERT, [56]) given by Eq (2), where $T_s$ and $T_f$ are the average times for successful and unsuccessful optimisations for that approach, and $\hat{p}$ is the fraction of the $n_{\text{Run}}$ optimisations that were successful for that approach.

$$ERT = T_s + T_f \frac{1 - \hat{p}}{\hat{p}} \tag{2}$$

The values of $T_s$ and $T_f$ are given in number of function evaluations. We assume that the total time spent on model simulations is proportional to the number of times the model is simulated. For a verification of this assumption, see Sect B in S1 Supporting Information and Figs A and B in S1 Supporting Information. We estimate the equivalent number of function evaluations when solving with sensitivities using the ratio of average times, across all model solves and approaches. These average times and the ratios are given for each problem in Table 3.

Table 3 shows large solve times with sensitivities for the Loewe I$_{\text{Kur}}$ and Moreno I$_{\text{Na}}$ problems, with time ratios going beyond what would be the case if finite differences were used to calculate the gradient. This is because these problems both use a cheap analytical solver for solves without sensitivities but require an ODE solver when solving with sensitivities in `myokit`. If finite differences were used for the Loewe I$_{\text{Kur}}$ and Moreno I$_{\text{Na}}$ problems, then the solves with sensitivities could be replicated by multiple solves without sensitivities and reduce the computation time. The time to emulate a solve with sensitivities through finite differences is also given in Table 3. When looking to convert gradient solves into the equivalent computational cost in solves without sensitivities, we use either the time to solve with sensitivities, or the hypothetical time to solve with finite differences, whichever is quicker. The time for a

**Table 3. Average solve times for the problems.**

| Problem | Cost time | Grad time | Grad time | Time ratio |
|---|---|---|---|---|
| | (without sensitivities; s) | (with sensitivities; s) | (finite differences; s) | (Grad/Cost; FEs) |
| Staircase HH | 0.00613 | 0.0537 | 0.0613 | 8.77 |
| Staircase MM | 0.00810 | 0.108 | 0.130 | 13.3 |
| Loewe $I_{Kr}$ | 0.00575 | 0.0604 | 0.0747 | 10.5 |
| Loewe $I_{Kur}$ | 0.00619 | 0.372 | 0.161 | 26.0 |
| Moreno $I_{Na}$ | 0.0575 | 2.53 | 0.977 | 17.0 |

Solve times are separated by solves with sensitivities (Grad time) and solves without sensitivities (Cost time). Reported times are averaged over all solves across all approaches. Grad time without sensitivities (using finite differences) is also reported as $(n_{Parameters} + 1)T_{cost}$. Time ratio reports how much more expensive gradient solves (either sensitivities or finite difference, whichever is quicker) are than cost solves. The SPSA algorithm used by Maryak1998 [20] is implemented in `ionBench` with the same solvers, but solves with sensitivities could use a finite difference method that only requires two solves without sensitivities, so the time ratio is set to 2 for this approach only.

gradient solve for each problem, expressed in units of time to solve without sensitivities (or function evaluations/FEs), is given in Table 3 as the 'Time ratio'.

We use ERT to rank all approaches. However, since ERT is a random variable (influenced by the randomly sampled initial parameters and any random effects inside the optimisation), whose variance in influenced heavily by the choice of $n_{Run}$, we need to ensure differences in ERT are sufficient to draw meaningful conclusions. We also need to consider the possibility that if no successes are observed for an approach, this does not mean the approach can be ignored. A sufficiently fast approach, which fails on the $n_{Run}$ parameters, may still be able to, on average, outperform approaches that saw successes if allowed to restart from more than $n_{Run}$ parameters. As such it is necessary to understand both when a difference in ERT is meaningful and unlikely to be due to random variation, and when a successful approach is likely to continue to outperform all failed approaches, even if the number of sampled parameters were to be increased.

To achieve this, we use bootstrapped hypothesis testing to compare the approach with the lowest ERT estimate against all other approaches. The choice of bootstrapping method is important here. A naive bootstrap would mean any approaches that failed on all $n_{Run}$ parameters would be guaranteed to only sample infinite ERTs (since $\hat{p} = 0$ regardless of the bootstrapped sample). An appropriate method should be able to approximate the uncertainty in the success rate when all observations are failures and quantify this when bootstrapping ERTs.

We begin with a simple bootstrapping algorithm. Of the $n_{Run}$ optimisations, we draw $n_{Run}$ samples with replacement, read whether or not each of these optimisation runs succeeded and how long they took (in function evaluations) and calculate the bootstrapped ERT estimate from these samples. When sampling from the times, if we have sampled only successes, the average failure time $T_f$ is given as a random sample from all times, reflecting the increased uncertainty in the failure time when all observed runs were successes. We similarly define the success time $T_s$ when all runs failed.

When sampling the success rate, we wish to impose two conditions. It should never be exactly zero (the next trial may succeed)—so that we derive only finite ERT estimates, and the success rate sampling should be smooth (there should be more than $n_{Run}$ discrete options). We do this by sampling from a Beta distribution ($a = x + 0.5$, $b = n_{Run} - x + 0.5$ where $x$ is the number of successes), which is the posterior distribution derived from the Jeffrey's prior (Beta(0.5, 0.5) for success rate) after observing $x$ successes from $n_{Run}$ attempts. This allows us to quantify uncertainty in the success rate, even if the finite number of observations we have taken are identical.

The Jeffrey's prior for the success rate was chosen since it is a standard uninformative prior which is not influenced by arbitrary parameterisations. For example, using a uniform prior on the success rate $p$ or a uniform prior on $p^2$ produce different posterior distributions, so while the intent of using a uniform distribution would be to provide no information, unintended information is provided through the parameterisation.

We perform draws of $10^4$ bootstrapped ERTs for each approach, and compare each sample from the best approach with each sample from each of the other approaches (a total of $10^8$ comparisons for each approach). As long as the best approach is better in 95% of comparisons, it is considered 'significantly better'. We term the proportion of samples where an approach is better than the best approach the approximate p-value, shortened to $\hat{p}$. We also check in case the best approach is worse in 95% of comparisons, in which case the other approach is considered 'significantly better'.

Pseudocode to generate these bootstrapped ERTs and determine significance is given in Algorithms A and B in S1 Supporting Information.

## Results

We begin by describing the parameter identifiability for each of the problems, derived from the profile likelihood plots. These plots inform the cost threshold used for each problem. Then we present the results of running all 34 unique approaches on the 5 test problems. This includes first evaluating the performance of approaches that observed at least one success, followed by evaluating the remaining approaches. Following this, we evaluate the significance of these results, including by running further optimisations where required. Finally, we utilise these results to inform the construction of a new approach, which is then evaluated and compared against the previous results.

### Parameter identifiability

The profile likelihood plots for each of the problems are reported in Figs D–I in S1 Supporting Information. The Staircase HH and Loewe $I_{Kr}$ problems showed good parameter identifiability across all parameters, as did the Moreno $I_{Na}$ problem. The Staircase MM showed much weaker parameter identifiability, with most parameters having very flat profile likelihood curves, where ODE solver noise begins to dominate the cost function surface. We also see a small bias in the global minimum introduced by the noise added to the simulated data, where the global minimum has shifted to parameters within approximately $\pm 20\%$ for the default values. The Loewe $I_{Kur}$ problem shows 11 parameters that were completely unidentifiable, where changes in one parameter could be fully compensated for by variation in other parameters (up to optimisation tolerances).

From these profile likelihood plots, the cost thresholds were generated and are reported in Table 4.

### Optimisation results

The estimated ERTs for the optimisation approaches that observed at least one success are given in Fig 6. The Wilhelms2012b approach (Trust Region Reflective with parameter bounds) [34], produced the lowest ERT estimate for all problems. While Wilhelms2012b did not provide the highest success rate (that goes to CMA-ES as in Beattie2018 [41], and the hybrid PSO/TRR and PSO/TRR+TRR methods as in Loewe2016d/e [3], depending on the problem), each run of Wilhelms2012b was significantly quicker — so that fast failures would allow more

**Table 4. Problem-specific cost thresholds.**

| Problem | Cost Threshold | Cost at true parameters | Noise |
|---|---|---|---|
| Staircase HH | 0.01570 | 0.01558 | iid noise added |
| Staircase MM | 0.005770 | 0.005767 | iid noise added |
| Loewe $I_{Kr}$ | $2.95 \times 10^{-6}$ | 0 | noise free |
| Loewe $I_{Kur}$ | $2.80 \times 10^{-7}$ | 0 | noise free |
| Moreno $I_{Na}$ | $1.70 \times 10^{-6}$ | 0 | noise free |

The cost thresholds that define a successful optimisation for each problem. The cost at the true parameters is presented for comparison.

initial guesses to be tried for the same budget, with a decrease in overall expected run time to find the global minimum.

It is important to note that results in Fig 6 are dependent on the time ratios given in Table 3 and other methods for calculating the gradient may give different time ratios. An equivalent figure with the solves with and without sensitivities separated is given in Fig C in S1 Supporting Information.

Fig 7 presents the full optimisation results. In Fig 7A, we have the best cost achieved by each approach (in its best optimisation run), with the approaches sorted by the time (in function evaluations) of this optimisation. Surprisingly, there is little correlation between the time the optimisation took and the cost achieved by the optimisation.

Fig 7B presents the time of the best cost, also sorted by this time. We see the vast majority of approaches take a similar amount of time, between $10^3$ and $10^5$ function evaluations, which likely comes from the convergence termination criteria at 2500 model solves (irrespective of the use of sensitivities). The approaches that produced the lowest times mostly struggled with optimisation, finding it difficult to suggest parameters inside the bounds, and then aborting early.

Finally, Fig 7C presents a plot of the best cost, sorted by the best cost. While not succeeding, some pairs did get close to their respective cost thresholds. The Staircase HH problem saw the Sachse2003b, Smirnov2020, Gurkiewicz2007b and Kohjitani2022 all get close to the cost threshold. Similarly, the Clausen2020, Jędrzejewski-Szmek2018, and Kohjitani2022 approaches all performed well on the Staircase MM problem, and the Beattie2018 and Dokos2004 on the Loewe $I_{Kur}$ problem. Of these approaches, the ones that stand out the most are Clausen2020, Dokos2004, Jędrzejewski-Szmek2018, and Beattie2018 which all succeeded on other problems.

Of all 170 approach-problem pairs, 25 (15%) failed to complete within the 1 week computation fixed time limit, or were otherwise incompatible (for example, the Cairns2017 approach [39] does not support negative parameters). For all approaches, we have complete data on at least two problems.

Reproductions of Fig 7 split by each problem (Figs J–N in S1 Supporting Information), and the raw numeric data (Table H in S1 Supporting Information) are provided in S1 Supporting Information.

## Significance

After performing the bootstrapping to verify the significance of the results, the Wilhelms2012b approach [34] was found to significantly outperform some approaches, but there are still many results that were not significant. This includes both: approaches that succeeded but were similar in performance to the Wilhelms2012b approach; and approaches that always failed but were significantly quicker than the Wilhelms2012b approach, where a future success

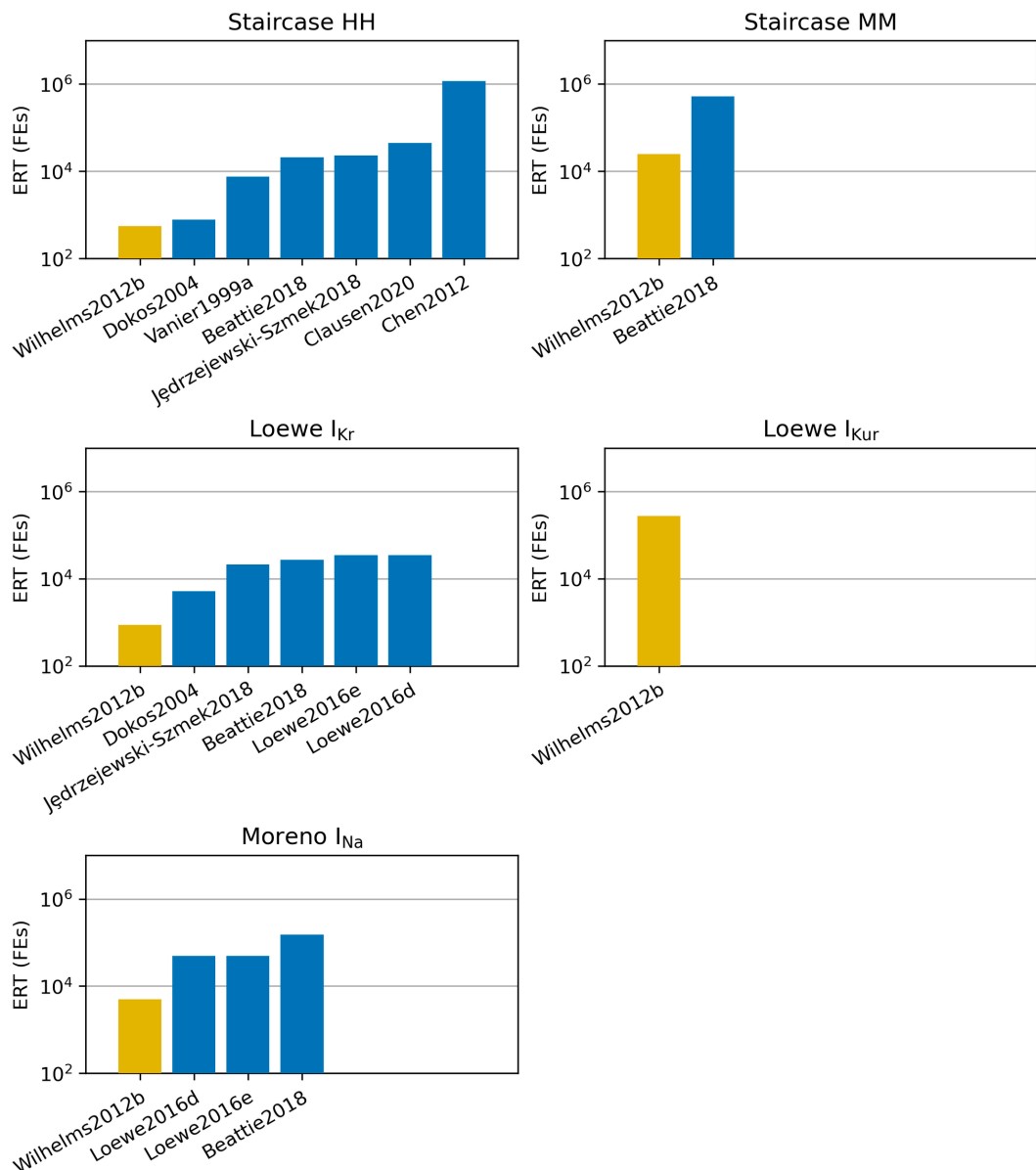

**Fig 6. ERT for successful approaches.** The expected run time (ERT) for each of the approaches that succeeded on at least one optimisation run, plotted for each of the problems, in units of function evaluations (FEs). Function evaluations with sensitivities are converted to their equivalent number of function evaluations without sensitivities using the time ratios from Table 3. The approach that gave the lowest ERT is shown in gold, while others are shown in blue.

may grant a lower ERT. The approaches that the Wilhelms2012b approach was not significantly better than are given in Table 5.

In some cases, larger $n_{\text{Run}}$ is required to ensure the Wilhelms2012b approach is actually the best choice. These approaches (shown in Table 5), and Wilhelms2012b, were rerun head-to-head until either Wilhelms2012b was significantly better or significantly worse. The pseudocode for this procedure (Algorithm C in S1 Supporting Information) and the number of runs required to show significance for each approach (Table G in S1 Supporting Information) are provided in S1 Supporting Information. Wilhelms2012b was found to be significantly

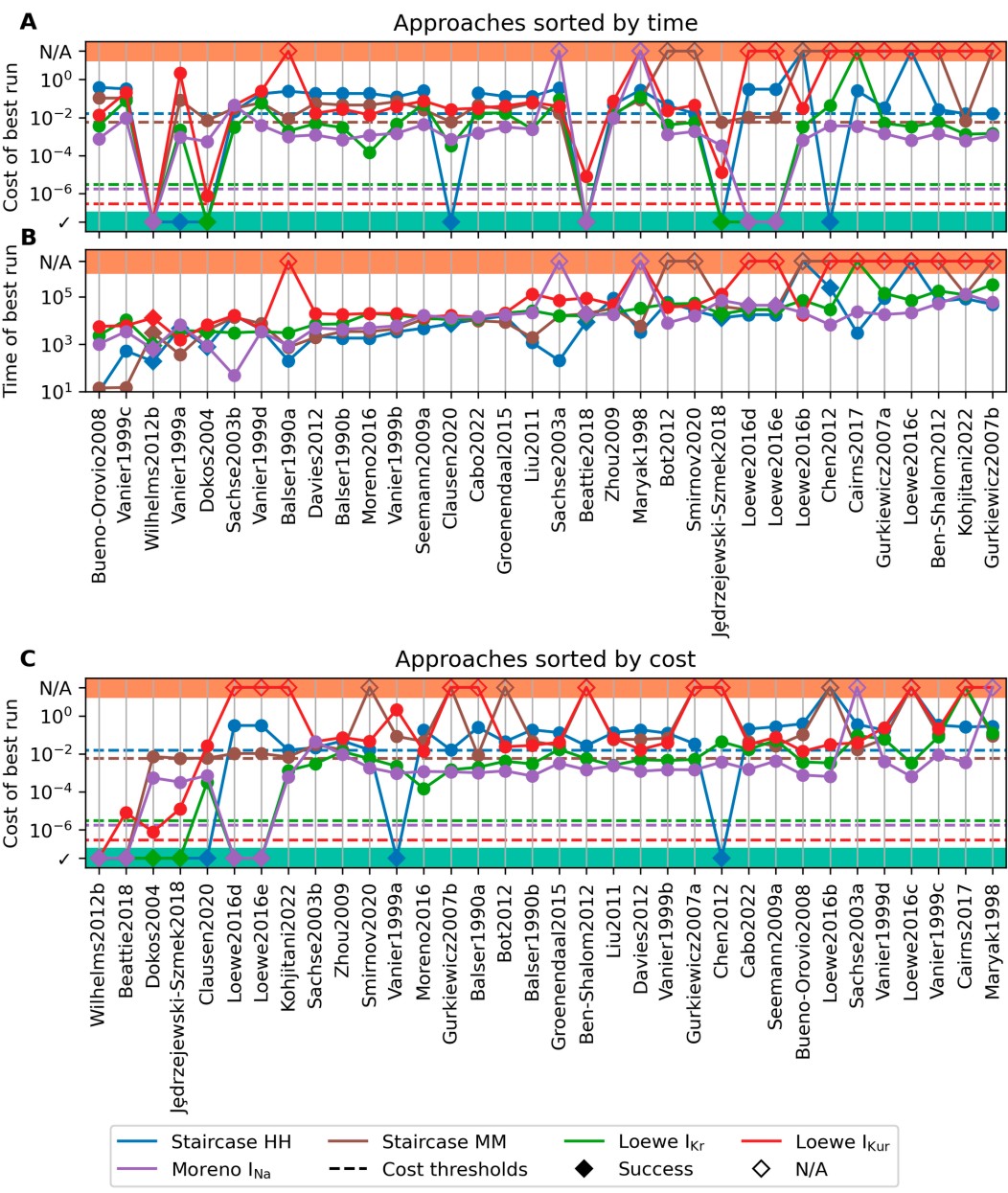

**Fig 7. Performance of all approaches.** Cost or time (in FEs) of the best run for each approach-problem pair, against approaches sorted (globally) by cost or time. (A) Cost, sorted by time. (B) Time, sorted by time. (C) Cost, sorted by cost. Cost thresholds are given as dashed lines. Successful runs are denoted by ✓; and incomplete runs by N/A.

better than almost all other approaches. The only exception to this was for the Staircase HH problem, where the Dokos2004 [23] approach was found to be significantly better.

## New approach

While we found the TRR (Trust Region Reflective) Wilhelms2012b approach [34] to generally be the most effective, the limited number of approaches may suggest better approaches can be found, as we only analysed those previously used in ion channel model optimisation. For

**Table 5. Bootstrapped significance.**

| Approach | Staircase HH | Staircase MM | Loewe $I_{Kr}$ | Loewe $I_{Kur}$ | Moreno $I_{Na}$ |
|---|---|---|---|---|---|
| Balser1990a | 0.11 | 0.47 | * | N/A | 0.20 |
| Balser1990b | * | 0.11 | * | * | 0.07 |
| Maryak1998 | * | * | * | N/A | N/A |
| Vanier1999a | * | 0.32 | * | 0.34 | 0.12 |
| Vanier1999b | * | 0.08 | * | * | 0.07 |
| Vanier1999c | * | 0.72 | * | 0.12 | 0.07 |
| Vanier1999d | * | 0.07 | * | 0.12 | 0.08 |
| Sachse2003a | * | 0.23 | * | * | N/A |
| Sachse2003b | * | 0.09 | * | 0.17 | 0.66 |
| Dokos2004 | 0.31 | 0.29 | * | * | 0.19 |
| Gurkiewicz2007a | * | N/A | * | N/A | * |
| Gurkiewicz2007b | * | N/A | * | N/A | * |
| Bueno-Orovio2008 | 0.57 | 0.81 | * | 0.21 | 0.23 |
| Seemann2009a | * | 0.06 | * | * | * |
| Zhou2009 | * | * | * | * | * |
| Liu2011 | * | 0.14 | * | * | * |
| Ben-Shalom2012 | * | N/A | * | N/A | * |
| Bot2012 | * | N/A | * | * | * |
| Chen2012 | * | N/A | * | N/A | 0.06 |
| Davies2012 | * | 0.11 | * | * | 0.09 |
| Groenendaal2015 | * | 0.06 | * | * | * |
| Loewe2016b | N/A | N/A | * | * | * |
| Loewe2016c | N/A | N/A | * | N/A | * |
| Loewe2016d | * | * | * | N/A | 0.13 |
| Loewe2016e | * | * | * | N/A | 0.13 |
| Moreno2016 | * | 0.11 | * | * | 0.07 |
| Cairns2017 | * | N/A | N/A | N/A | * |
| Jędrzejewski-Szmek2018 | * | * | * | * | * |
| Beattie2018 | * | 0.09 | * | * | 0.07 |
| Clausen2020 | * | * | * | * | * |
| Smirnov2020 | * | N/A | * | * | * |
| Cabo2022 | * | * | * | * | * |
| Kohjitani2022 | * | * | * | N/A | * |

The bootstrapped significance for each approach-problem pair. Duplicate approaches (see Table 1) are not reported since their results are identical to a reported approach. * means that the Wilhelms2012b approach was significantly better. If the Wilhelms2012b approach was not significantly better, then the approximate p-value $\hat{p}$ is shown. N/A means the approach-problem pair failed to complete the full $n_{Run}$ optimisations in the 1 week limit, except for Cairns2017 which requires parameters to be positive so is incompatible with some problems.

example, log transforms have been previously shown to be effective at improving optimisation for CMA-ES in ion channels [12] and in other areas of systems biology [6,57,58], but TRR with log transforms was not evaluated in the above results since it is not a previously used approach. The same is true for rate bounds, where reducing the size of the parameter search can improve optimisation.

We derived a new approach, utlising TRR as the optimiser and parameter bounds, as in Wilhelms2012b [34], but also included rate bounds and log transforms.

This new approach was evaluated against the previous best approach for each problem (Dokos2004 for Staircase HH, Wilhelms2012b for others). The approaches were run on the same parameters as each other, until the difference in ERT was significant (Staircase HH: 500, Staircase MM: 400, Loewe $I_{Kr}$: 100, Loewe $I_{Kur}$: 250, Moreno $I_{Na}$: 3000). The ERT for each approach-problem pair is given in Fig 8.

Using the new approach provides between a 21% and 84% reduction in the ERT compared with the previous best approach for each problem.

## Discussion

It is challenging to choose an optimisation approach to use *a priori*. Ion channel models present many conflicting complexities in optimisation that influence such a choice (parameter unidentifiability, noisy data, both fast and slow timescales, high numbers of data points, reasonably large numbers of parameters to optimise). Making a poor choice leads to a waste of computational resources and time at best, but may lead to sub-optimal identification of parameters, limiting the usability and predictive power of the final model. We believe comparing many optimisation approaches against a range of literature problems is the best way to decide which optimisation approach to use for these types of problems.

We found the Wilhelms2012b approach consistently performed well, however, it was not the best across all previously applied approaches. For the Staircase HH problem, the Dokos2004 approach performed significantly better. The TRR optimiser from Wilhelms2012b searches in a weighted-average direction between the steepest descent and Gauss-Newton directions. Similarly, the curvilinear gradient descent optimiser from Dokos2004 searches in a curved path starting in the direction of steepest descent and terminating at the Gauss-Newton step. That both these approaches performed well indicates Gauss-Newton steps seem very beneficial in ion channel optimisation, which in turn suggests that the cost function surfaces are likely highly parabolic. While the radius of convergence of the global minimum for Gauss-Newton steps may be small in some problems (such as Loewe $I_{Kur}$), being able to efficiently find local minimums means the optimiser can terminate early and repeat from a new starting location.

While we did observe that the majority of approaches did not succeed in any of the runs, that does not mean they will never succeed. When evaluating these approaches and allowing multiple restarts, it is worth remembering that all approaches will eventually succeed after enough attempts. It may also be the case, that for a more/less strict definition of success, a different approach may be more efficient. In many cases, where these approaches are evaluated

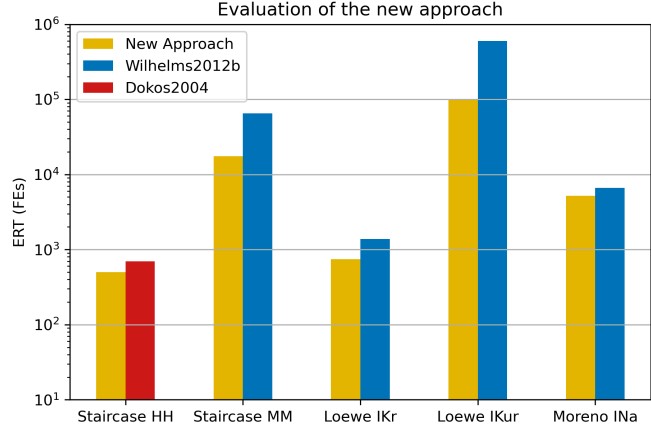

**Fig 8. New approach.** Comparison of ERT between the new approach and the previous best approach for each problem. In all cases, the new approach is significantly better than the previous best.

against synthetic data, a clear definition is success is not given. Rather performance is measured by whether or not the data produced after optimisation appears to reproduce the synthetic data used to optimise against in a single run. The fact that so many approaches are able to meet this requirement, while also struggling to identify costs closer to the global minimum demonstrates a need for stricter evaluation of optimisation approaches.

Approaches that run for shorter lengths of time may also indirectly receive information on the parameter bounds through the sampling function. Since initial parameters for an optimisation are always within the bounds, an optimiser that does not move as far (and favours restarting quickly) is less likely to suggest points outside of the bounds, and therefore may see some small indirect benefit over longer approaches (if both approaches do not use bounds).

Additionally, there may be benefits to be found in methods outside of the current scope of `ionBench`. Changes to the representation of the data [59] and varying the size of the bounds over the optimisation [42] have been previously shown to be beneficial in some instances.

Through defining a new approach, we sought improvements upon both the Wilhelms2012b and Dokos2004 approaches. This new approach was derived from a combination of the results presented here, and other works investigating the effect of different modifications to ion channel or general systems biology optimisations, and significantly outperformed the previously applied approaches across all problems.

TRR has been found to perform well in other systems biology optimisation benchmarks [58]. The use of cheap local optimisers with multi-start or hybrid approaches has also been shown to be effective in other areas of systems biology [60]. There may be further improvements to be found by introducing a smarter restart strategy that utilises a global meta-heuristic to choose new parameters based on the previous optimisations, rather than relying on the parameter sampling function [4,60].

When utilising a multi-start approach, it is necessary to consider how many restarts may be required. Success rates for the Wilhelms2012b approach were between 2% and 70%, so using 100-200 restarts seems like a reasonable choice. Since it is not possible to identify whether or not a run was successful in practice, verifying whether or not the number of restarts was sufficient can be a challenging problem. One possible method to verify that the number of restarts was likely sufficient is to check that the best performing points all ended up in the same minimum. This can be done by plotting the cost between those points and verifying that the surface has no local maxima. Alternatively, optimisation can be restarted until the best cost has not been improved upon for around 100 restarts.

The choice of model solver is important when considering the best optimiser for a given problem. The present work makes use of CVODES sensitivities to calculate the gradient, but in some cases, where analytical solutions are tractable, this may not be the best choice. Notably, the method by which the model is solved, both with and without sensitivities, alters how costly gradient computations are, and therefore can skew the optimisation times, either in favour or against, approaches that make use of gradient information. In situations where sensitivity solvers are not viable, and a finite difference approximation for the gradient is insufficient, CMA-ES and the Beattie2018 approach may be the best choice. It is the only approach to succeed in four of the five problems without using gradient information, and for the remaining problem (Loewe $I_{Kur}$), it performed second best in cost minimisation, only behind the Wilhelms2012b approach.

We have not considered hyperparameter optimisation in this work. This decision has been recommended previously [4], as the choice of hyperparameters is typically problem specific and optimising them can be costly, potentially outweighing the benefits achieved by the use of optimised hyperparameters. We have attempted to use reasonable *a priori* choices for the hyperparameters based on literature recommendations. However, some of the approaches are

described without recommendations on hyperparameters. Providing recommended hyperparameter values, or recommendations on how to choose reasonable hyperparameter values *a priori*, would improve the usability and real-world efficiency of any newly developed approaches.

Emulators, machine learning models which approximate model outputs for a given set of parameters, are becoming an increasingly useful tool during parameter optimization. The results here are generally applicable regardless of the method used to identify the cost of the parameters, where calculating the cost through an emulator would require a similar number of function evaluations (ERT), but may be performed faster in real world time due to the reduced time to evaluate the cost. However, they may differ in the calculation of gradients. Depending on the emulator used, gradients may or may not be calculated quickly (with a different time ratio to that seen for solving the ODEs), and may or may not be accurate estimates of the gradient of the true cost function. When using emulators, it may be beneficial to use a method which does not rely on gradients such as CMA-ES and the Beattie2018 approach.

Parallelisation of code is another common technique for reducing the computational time. In some cases, the ability to easily parallelise the optimisation routine (such as for particle-based or genetic algorithms) may be a good reason to choose one approach over another. However, most optimisations should be repeated from differing initial guesses anyway [12], and running independent optimisations from different initial guesses is an 'embarrassingly parallel' task (trivial to parallelise, perfectly scaling with the number of available processors). This is particularly true for gradient-descent based optimisers, such as TRR, where the multi-start strategy is crucial.

In future, exploration into a wider range of optimisation problems including those that deal with real data and model discrepancy could prove useful. `ionBench` is not currently set up to use real data, as the assessment of optimisation runs (whether they succeeded or failed) is determined using the cost around some 'true' (synthetic data generating) parameters, which would not be available in these applications—however using the best score and parameter set that any optimiser ever found would be an alternative.

Including additional problems that explore how the strength of the noise alters optimisation performance would also be beneficial. Noise can act to smooth the cost function surface, reducing the number of local minima and making it easier to find the global minima, particularly for local approaches like gradient descent [59] (although this new global minima may not correspond to the data generating parameters for large amounts of noise). A wider exploration, with a range of problems of varying noise levels may favour different approaches. Now that a smaller subset of the approaches have been shown to perform well, these approaches could be more thoroughly tested on a wide range of problems, without the computational cost of running all thirty-four approaches against each problem.

While we have focussed on the choice of optimisation approach, it is not the only way to improve optimisation. There are specific choices in the construction of the problem that can improve the quality of the optimisation; such as using current traces of varied or complex voltage protocols over simplified summary statistics [12], or model reduction when parameters are unidentifiable [61]. Simply put, ion channel optimisation should consist of both a robust and carefully constructed problem and an efficient optimisation approach. This could also include parameter transforms (for both the initial-guess sampling function and optimiser parameter-space to model parameter-space mapping), which can dramatically improve the performance of optimisers [11,12].

This work focussed on cost function minimization, however, the true goal is to identify the data generating parameters. Unfortunately, assessing approaches on parameter error can present some problems. Since the inclusion of noise can warp the cost function surface,

including changing the parameters that produce the global optimum, the goal of cost function minimisation can become misaligned with the goal of minimising parameter error. This can be problematic since the best cost achieved during an optimisation is not necessarily the one that was closest to the data generating parameters. Making changes to the problem, as described above, can help also help to better align these goals. Utilising a more informative voltage protocol can help to better identify parameters, which can reduce the size of the perturbation seen when adding noise. Similarly, model transformations such as model reduction can reduce model sloppiness, which can also reduce the size of this perturbation.

We have made `ionBench` available, open source, to allow future approaches to be evaluated against its test suite, and future tests to be added. We hope this will assist in both the development of new efficient approaches and provide clear comparisons against existing works.

In conclusion, we have compared a wide range of different optimisation approaches against a test suite of cardiac ion channel optimisation problems. We found a new approach, derived from the Wilhelms2012b approach, to be the most efficient. As part of a multi-start strategy, it can successfully optimise parameters at significantly reduced computational time compared with other approaches. This improvement can be leveraged to aid where parameter optimisation presents as a particularly computationally demanding task, such as in the identification of model structures [62], training model populations for assessing model discrepancy [63], and optimal experimental design [64].

## Acknowledgments

We are grateful for access to the University of Nottingham's 'Ada' high performance computing service.

## Supporting information

**S1 Text. A: Problem definitions.** A detailed description of the problems implemented in ion-Bench, including the protocols, data, parameter sampling methods and simulation settings. **B: Model solve time.** A comparison of model solve times for different approaches. **C: Significance.** Pseudocode implementations of the ERT bootstrapping algorithm and further method details and results for testing significance. **D: Profile likelihood.** Profile likelihood plots for each of the five problems. **E: Performance of all approaches.** Reproductions of Fig 7 showing the performance of all approach-problem pairings on their best run.
(PDF)

## Author contributions

**Conceptualization:** Matt J. Owen, Gary R. Mirams.

**Data curation:** Matt J. Owen.

**Formal analysis:** Matt J. Owen.

**Funding acquisition:** Gary R. Mirams.

**Investigation:** Matt J. Owen.

**Methodology:** Matt J. Owen, Gary R. Mirams.

**Project administration:** Gary R. Mirams.

**Software:** Matt J. Owen.

**Supervision:** Gary R. Mirams.

**Visualization:** Matt J. Owen, Gary R. Mirams.

**Writing – original draft:** Matt J. Owen.

**Writing – review & editing:** Matt J. Owen, Gary R. Mirams.

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
