## [Decision Letter · Decision Letter 0]

8 May 2025

PCOMPBIOL-D-25-00198

IonBench: a benchmark of optimisation strategies for mathematical models of ion channel currents

PLOS Computational Biology

Dear Dr. Mirams,

Thank you for submitting your manuscript to PLOS Computational Biology. After careful consideration, we feel that it has merit but does not fully meet PLOS Computational Biology's publication criteria as it currently stands. Therefore, we invite you to submit a revised version of the manuscript that addresses the points raised during the review process.

Please submit your revised manuscript within 30 days Jul 08 2025 11:59PM. If you will need more time than this to complete your revisions, please reply to this message or contact the journal office at ploscompbiol@plos.org. Please include the following items when submitting your revised manuscript:

We look forward to receiving your revised manuscript.

Kind regards,

Frédéric E. Theunissen

Academic Editor

PLOS Computational Biology

Stacey Finley

Section Editor

PLOS Computational Biology

**Additional Editor Comments :**

Dear Authors,

The two reviewers that were assigned to your manuscript agree that it is an appropriate contribution for Plos Comp Biol. However, they raised some minor to less-minor issues that I would like you to address.

Thank you.

Frederic Theunissen

**Journal Requirements:**

3) We notice that your supplementary Figures, Tables, and information are included in the manuscript file. Please remove them and upload them with the file type 'Supporting Information'. Please ensure that each Supporting Information file has a legend listed in the manuscript after the references list.

4) Please amend your detailed Financial Disclosure statement. This is published with the article. It must therefore be completed in full sentences and contain the exact wording you wish to be published.

5) Please provide a completed 'Competing Interests' statement, including any COIs declared by your co-authors. If you have no competing interests to declare, please state "The authors have declared that no competing interests exist". Otherwise please declare all competing interests beginning with the statement "I have read the journal's policy and the authors of this manuscript have the following competing interests:"

**Reviewers' comments:**

Reviewer's Responses to Questions

Reviewer #1: Owen & Mirams present a benchmark study on optimization approaches for parameters in cardiac ion channel models, typically described by several coupled ordinary differential equations. This is an original contribution and will be of high importance to researchers in the field, especially given the well-documented and openly available problem definitions. The authors invested great effort in implementing 34 approaches described in literature to rigorously compare them and describe the methodology clearly.

Based on the results of their benchmark, which provides methodological insight, they propose an innovative adaption of existing approaches that consistently performs best on their set of test problems. All data and code are available with documentation and tests.

I have the following comments that could further increase the value of this study_

- Lines 157ff: The choice of benchmark problems is not obvious:

- What's the rationale behind choosing those 4 specific problems?

- Why were problems 1-4 evaluated by comparing currents whereas 5 was compared in terms of summary statistics? (line 176)

- Noise was added to benchmark problems 1+2 but not to 3-5. Why? Would you suspect the results to change if noise was present (as is inevitably the case in any real-world application of these methods)? Previous studies reported that small levels of noise can even lead to better results for gradient-based optimization whereas performance gets worse for higher noise levels.

- The versatility of the provided code could be increased by exposing the level of noise as a configurable parameter.

- Why are the parameter sampling regions for problem 5 smaller (+/-25%) than for the other problems?

- It would be interesting to see the cost values and runtime achieved with all approaches (even if not undershooting the cost threshold considered "successful") as the small number of approaches (compared to the 34 candidates) shown in Fig. 6 is surprising.

- The suggested optimal algorithm is a multi-start approach. Some guidance on the number of required starts would be helpful for future users of this approach as the achievable cost in real-world scenarios will be markedly above 0.

Further points:

- The relation of the described approaches with machine learning approaches for emulating the AP and for parameter optimization could be further elaborated.

- Fig. 2: Should "RMSE: cost" read "RMSE, cost?"

- Fig. 5 and lines 255ff: I have trouble identifying the lower bound 10^-3 in the figure. Please elaborate the caption or adapt the figure.

- Lines 297ff./404: I consider the information on the unidentifiable parameters very valuable. Please provide more guidance on how to read and interpret the plot in the supplement and consider moving key results (which ones were unidentifiable?) to the main manuscript

- Fig. 7 is a bit hard to read with intersecting lines and overlapping symbols. How about providing this information in addition in a supplementary table?

Reviewer #2: Review of article PCOMPBIOL-D-25-00198

The manuscript discusses a developed open source software suite for evaluating optimization methods for inverting model parameters, using data generated from a number of ion channel mathematical models. The authors use this software to benchmark a wide range of optimization approaches from the literature for these chosen model formulations, and from the results, put forth an improved methodology with greater performance in their test suite. The article is clear, well written, and descriptive, and adds a useful tool for researchers working in this space. It is largely appropriate for publication as written.

There are a few comments I would ask the authors to address though, to help add clarity to the results and conclusions.

1. It was somewhat surprising that so many (the vast majority) of the published methods were unable to effectively minimize the cost for any of the chosen formulations. I would ask for some discussion if this finding is consistent with the cited publications on these methods, or if it is a function of the methods chosen in their developed benchmark system? In other words, were these published methods able to solve similar problems elsewhere, but not in your benchmark?

2. The choice of model set-up could be better discussed, in particular the choice of noise. Why was this specific level of noise added, and why to only a few of the model formulations? The choice seems a little arbitrary, and yet could potentially be having a strong effect on the results.

3. All benchmarks are shown in terms of cost and time. I think the results would be well supplemented with additionally showing parameter error, as this in the end is the actual goal of this type of optimization process. This obviously could show poor results in the case where there are so many parameters that are unidentifiable, but I think the reader can contextualize it sufficiently.

4. In the profile curves, I was uncertain of the difference between the ‘optimized’ and ‘unoptimized’ profiles.

**Have the authors made all data and (if applicable) computational code underlying the findings in their manuscript fully available?**

Reviewer #1: Yes

Reviewer #2: Yes

PLOS authors have the option to publish the peer review history of their article (what does this mean?). If published, this will include your full peer review and any attached files.

Reviewer #1: No

Reviewer #2: No

**Figure resubmission:**
---

## [Editor Report · Decision Letter 1]

9 Jul 2025

Dear Professor Mirams,

We are pleased to inform you that your manuscript 'IonBench: a benchmark of optimisation strategies for mathematical models of ion channel currents' has been provisionally accepted for publication in PLOS Computational Biology.

Best regards,

Frédéric E. Theunissen

Academic Editor

PLOS Computational Biology

Stacey Finley

Section Editor

PLOS Computational Biology

Dear Matt and Gary,

Thank you for addressing all the reviewers comments. I found your thorough review of published optimization methods and your description of the strengths and weaknesses very useful. I hope that your paper becomes a reference for all researchers modeling ion channels.

Best,

Frederic Theunissen

---

## [Editor Report · Acceptance letter]

PCOMPBIOL-D-25-00198R1

IonBench: a benchmark of optimisation strategies for mathematical models of ion channel currents

Dear Dr Mirams,

I am pleased to inform you that your manuscript has been formally accepted for publication in PLOS Computational Biology. Your manuscript is now with our production department and you will be notified of the publication date in due course.

With kind regards,

Anita Estes
